# Effects of Soybean and Linseed Oils Calcium Salts and Starter Protein Content on Growth Performance, Immune Response, and Nitrogen Utilization Efficiency in Holstein Dairy Calves

**DOI:** 10.3390/ani13060960

**Published:** 2023-03-07

**Authors:** Ardashir Rajabi, Farshid Fattahnia, Mohammad Shamsollahi, Hossein Jahani-Azizabadi, Hamed Khalilvandi-Behroozyar, Adel Pezeshki, Mehdi Kazemi-Bonchenari

**Affiliations:** 1Department of Animal Science, Faculty of Agriculture and Natural Resources, Ilam University, Ilam 69315-516, Iran; 2Department of Animal Science, Faculty of Agriculture, Kurdistan University, Sanandaj 6617715175, Iran; 3Department of Animal Science, Urmia University, Urmia 5756151818, Iran; 4Department of Animal and Food Sciences, Oklahoma State University, Stillwater, OK 74078, USA; 5Department of Animal Science, Faculty of Agriculture and Natural Resources, Arak University, Arak 38156-8-8349, Iran

**Keywords:** dairy calf, starter feed, n-6 fatty acid, n-3 fatty acid, crude protein content, immunity, performance

## Abstract

**Simple Summary:**

In the current study we evaluated two fatty acid sources (soybean oil contained n-6 FA vs. linseed oil contained n-3 FA) along with two different starter protein contents (18 vs. 22% CP). Our findings show that different FAs can impact nitrogen utilization efficiency. Moreover, the immune system of calves is influenced by the interaction of FAs and starter protein content. This is the first report on the interaction effect of FAs and nitrogen content of starters in dairy calves, which can be interesting for both farmers and researchers in order to have optimized starter formulation in terms of the fatty acid source and starter protein contents in an accelerated growth program for calves in commercial dairy farms.

**Abstract:**

This study aimed to investigate the interaction of fatty acid (FA) source [calcium salt of soybean oil (n-6 FA) vs. calcium salt of linseed oil (n-3 FA) both 3% DM basis] with protein content (18% vs. 22% CP, based on DM) on growth performance, blood metabolites, immune function, skeletal growth indices, urinary purine derivatives (PD), and microbial protein synthesis (MPS) in young dairy calves. Forty 3-day-old calves (20 females and 20 males) with a starting body weight (BW) of 40.2 kg were assigned in a completely randomized block design in a 2 × 2 factorial arrangement of treatments. Experimental diets were: (1) n-6 FA with 18% CP (n-6-18CP), (2) n-6 FA with 22% CP (n-6-22CP), (3) n-3 FA with 18% CP (n-3-18CP), and (4) n-3 FA with 22% CP (n-3-22CP). Starter feed intake and average daily gain (ADG) were not influenced by experimental diets (*p* > 0.05). However, before weaning and the entire period, feed efficiency (FE) was greater in calves fed n-3 FA compared to n-6 FA (*p* < 0.05). Heart girth (weaning, *p* < 0.05) and hip height (weaning, *p* < 0.05 and final, *p* < 0.01) were highest among experimental treatments in calves who received n-3-22CP diets. The greatest blood glucose (*p* < 0.05) and insulin (*p* < 0.01) concentrations in the pre-weaning period and the lowest serum concentration of tumor necrosis factor (before weaning, *p* < 0.05) were observed in calves fed the n-3-22CP diet. However, the greatest blood urea N (before weaning, *p* < 0.05; after weaning, *p* < 0.05) and urinary N excretion (*p* < 0.05) were found in calves fed n-6-22CP diets compared to other experimental arrangements. In conclusion, offering calves with Ca-salt of n-3 FA along with 22% CP content may be related to improved nitrogen efficiency and immune function.

## 1. Introduction

Supplementation of fat has been proposed as a practical strategy in commercial dairy production programs for increasing the energy concentration of starter diets in young animals. This would be a valuable strategy for supplying the high energy demand of young calves because of the high growth rate in this period of the lifecycle, and, conversely, the low starter intake to supply this high demand [1]. Fat supplementation has potential to provide a higher energy level per unit of feed consumed [2]; thus, can compensate, to some extent, for the high energy requirement of young calves that have limited feed intake in the early stages of growth. Lower nutrient digestibility [2,3], lower starter feed intake [4], looser feces [5], lower microbial activity, and less microbial protein synthesized in the rumen [6] have been previously reported as negative aspects of supplemental fat in young calves. Fat saponification with calcium (Ca) has been practiced to solve some of the abovementioned problems and is mainly due to bypassing the fatty acids (FA) to the small intestine [7]. However, because of partial dissociation of the calcium salts in the rumen, partial negative effects of this method still remain [8]. Regardless of the passage rate of Ca-soaped FA from the rumen, some negative impacts of FAs on animal performance can also be related to the FA source [9,10]. Thus, one of the goals of the current study was to compare the Ca salts of n-6 (soybean oil) and n-3 (linseed oil) in pre-weaning dairy calves.

Previous reports indicated that supplemental fat reduces the metabolizable protein (MP) reaching to small intestine [4]. This probably is due to lower crude protein (CP) digestibility [11], and consequently lower microbial protein synthesis (MPS) within the rumen of animals fed high-fat diets [4]. Further, it has been indicated that supplementation of certain individual FAs may improve the efficiency of nitrogen (N) in dairy cows [12], young goat kids [13], and young lambs [14]. Some researchers believe that the higher N efficiency while supplementing with special FA can be related to the influence of these FAs on gene expression level [15], or through hormonal regulation [12]. Limited publications are available on the influence of individual FAs (either n-6 or n-3 FA) on N efficiency in dairy calves. In the present experiment, we evaluated the effect of Ca salt of n-6 FAs supplied through soybean oil (SBO) and Ca salt of n-3 FA supplied through linseed oil (LSO) on N metabolism in calves fed starter with either 18 or 22% CP content during the pre-weaning period.

## 2. Materials and Methods

The current study was conducted in a commercial dairy farm (Bawan Kalhor Agriculture and Animal Husbandry, Kermanshah, Iran) from 20 November 2021 until 25 January 2022. All calves were reared and managed according to the guidelines of the Iranian Council of Animal Care (ICAC, 1995).

### 2.1. Calves, Experimental Diets, and Management

A total of forty (20 males and 20 females) healthy Holstein dairy calves were selected at 3 d of age and randomly allocated to 4 groups (10 calves per treatment; 5 males and 5 females per each) in a 2-by-2 factorial arrangement. The starter FA sources and CP content were Ca-salt of soybean oil as n-6 FA source vs. Ca-salt of linseed oil as n-3 FA source and 18% vs. 22% (DM basis), respectively. Immediately after birth, each calf’s body weight was recorded and they were housed in 1.2 × 2.5 m individual pens. The floor of each pen was bedded with sand and renewed every 24 h. Each calf was fed 5 L of colostrum during the first 12 h of life (2.5 L within 1.5 h after birth and 2.5 L in a second feeding), which was continued during the first 2 days of life. Colostrum quality was evaluated using a digital Brix refractometer (PAL-1, Atago Co. Ltd., Bellevue, WA, USA) and only colostrum with >22 on the Brix scale was used in this study. Calves were fed 4.5, 7, 2, and 1 L/d of whole milk from d 3 to 7, 11 to 40, 41 to 60, and 61 to 62, respectively. Fat, protein, lactose, and total solid of milk samples were measured at weekly intervals using an infrared spectrophotometer (FOSS Milk-O-Scan, FOSS Electric, Hillerod, Denmark). The average composition of consumed milk was 31.4 ± 0.8, 31.0 ± 0.4, 48.9 ± 0.4, and 119 ± 0.2 g/kg of fat, protein, lactose, and total solid, respectively. All calves were weaned at 63 d of age but maintained on the experimental diets until 83 d of age. Experimental diets were formulated according to NRC [16] recommendations to meet nutrient requirements, which differed in FA source and CP content. Thus, 4 experimental diets were evaluated: (1) starter diet with 18% CP and 3% Ca-salt of soybean oil as n-6 FA source (n-6-18CP); (2) starter diet with 22% CP and 3% Ca-salt of soybean oil as n-6 FA source (n-6-22CP); (3) starter diet with 3% Ca-salt of linseed oil as n-3 FA source and 18% CP (n-3-18CP); and (4) starter diet with 3% Ca-salt of linseed oil as n-3 FA source and 22% CP (n-3-22CP). The feed ingredients and the chemical composition of experimental starter diets are given in Table 1. Ca-salt of soybean oil (PersiaFat-Omega-6) and Ca-salt of linseed oil (PersiaLin-Omega-3) were provided by Kimia Danesh Alvand Co. (Qom, Iran) which contained 51.2% linoleic acid (n-6) and 42% linolenic acid (n-3), respectively. All experimental calves had free access to starter diets and water throughout the experiment.

### 2.2. Dry Matter Intake, Weight Gain, and Feed Efficiency

Experimental diets were offered once daily at 0800 h and orts were collected and recorded at next day morning (0730 h). The starter DM intake for each calf throughout the study was calculated based on the amount of offered starter feed and ort. Calves’ BWs were measured immediately after birth (d 3) and then every 10 d during the study, which was recorded before feeding to avoid having the effects of gastrointestinal tract content. Average daily gain (ADG) and feed efficiency (FE: kg of BW gain divided by kg of total DM intake which is the summation of milk DM and starter DM) before (d 3–63) and after (d 64–83) weaning, and entire periods (d 3–83), were also calculated. Samples of experimental diets and orts were dried at 60 °C for 48 h in a convection oven every 14 d. To determine chemical composition, subsamples of dried experimental starter diets and their orts were mixed thoroughly and ground in a mill (Ogaw Seiki CO., Ltd., Tokyo, Japan) to pass a 1-mm screen. Experimental diets and their orts were analyzed for DM (method, 2001.12), ash (method No. 942.05), CP (method No. 991.20), and ether extract (EE; method No. 920.39) content according to standard procedures [17]. To determine neutral detergent fiber (NDF), samples of experimental diets were analyzed by applying a heat-stable alpha-amylase with no use of sodium sulfite addition and correction for residual ash and N [18].

### 2.3. Skeletal Growth Parameters

Skeletal growth parameters of all calves including wither height, hip height, heart girth, and body length were measured at the beginning of the experiment (d 3), at weaning (d 63), and at the last day of the experiment (d 83).

### 2.4. Blood Items, Insulin, and Inflammatory Indicators

Blood samples were obtained without preservative on d 36 (before weaning) and 72 (after weaning) from the jugular vein of the calves 2 h after the morning feeding and were allowed to clot. Immediately after centrifugation of clotted samples at 3000 g for 15 min at 4 °C, serum was separated and stored at −20 °C until subsequent measurements. Serum concentrations of glucose (kit no. 93008) and urea N (BUN; kit no. 93013) were determined using an autoanalyzer (UNICCO, 2100; Zistchemi Co., Tehran, Iran) and analytical kits (Pars-Azemun Co. Ltd., Karaj, Iran). Serum beta-Hydroxybutyrate (BHB) and insulin concentrations were measured using a commercial kit (Abbott Diabetes Care Ltd. Alameda, CA, USA) and ELISA (Auto Analyzer Hitachi 717), respectively. The concentration of inflammatory markers including serum amyloid A (SAA, Bioassay, Shanghai, China) and tumor necrosis factor (TNF-α) were determined using ELISA kits (Karmania Pars Gene, Iran) [19]. The haptoglobin (HP) concentration was measured using immunoturbidimetric assay (Biorex-fars, Shiraz, Iran) ELISA kits (DANA 3200, Garny, Iran).

### 2.5. Urinary PD, MPS, and Urinary N Excretion

A spot urine sampling procedure was used to estimate microbial protein synthesis (MPS) in the rumen, which is based on the purine derivatives (PD) excretion in urine. Because it has been stated that bovine milk containing PD interferes with MPS in suckling calves [20], urinary PD was measured only in post-weaned calves. Urinary creatinine excretion was used to estimate total urine excretion by using the model described here; BW × 26.8/creatinine concentration (mg/L) [21]. Urine spot samples (approximately 10 mL) were collected on three consecutive days during the post-weaning period (between 0900 and 1100 h and 1500 and 1700 h) at the time that calves had spontaneous urination. A sub-sample of 5 mL of each sample was diluted immediately with 45 mL of 0.036 N sulfuric acid and stored at –20 °C for analysis. The concentrations of uric acid, creatinine, and allantoin were measured in sub-samples as extensively explained in previous work [22]. Urinary N (UN) content was measured according to the assay described previously [23]. Estimated daily urine output was used to calculate daily urinary excretion of allantoin, and uric acid as total daily PD. The ruminal MPS was calculated from total daily PD output [24].

### 2.6. Statistical Analysis

The collected data were analyzed using PROC MIXED of SAS (version 9.1; SAS Institute, Cary, NC, USA). An individual calf was used as the experimental unit. Repeated measure was applied for analyzing the starter DM intake, ADG, and FE with period (each period was 10 d) as the repeated variable. The following model was used: Yijkl = μ + FAi + CPj + Tk + (FA × T)ik + (CP × T)jk + (FA × CP)ij + (FA × CP × T)ijk + β(Xi − $\bar{X}$) + εijkl, where Yijk is the dependent variable; µ is the overall mean; FAi is the impact of FA source (calcium salt of n-6 FA vs. calcium slat of n-3 FA); CPj is the effect of starter CP content (18% vs. 22%, DM basis); Tk is the effect of period; (FA × T)ij is the interaction between FA source and period; (CP × T)ik is the interaction between starter CP content and period; (FA × CP)jk is the interaction between FA source and starter CP content; (FA × CP × T)ijk is the tripartite effect of FA source, starter CP content, and period; β(Xi − $\bar{X}$) is considered as covariate and εijk is the overall error term. The sex effect was not significant in the current study. An autoregressive (order 1) covariance structure was chosen according to the Akaike and Bayesian information criteria. Data were analyzed for 3 periods including before weaning (d 3–63), after weaning (d 64–83), and the entire period (d 3–83). The initial measurements (d 3) were included as a covariate for the analysis of BW and skeletal growth parameters at weaning and at the end of the experiment. Tukey’s multiple range tests were used to determine the differences among experimental diets’ means. With respect to significance, the obtained results have been considered significant when *p* ≤ 0.05, and a tendency was considered when 0.05 < *p* ≤ 0.10. All reported values are considered as least square means.

## 3. Results

### 3.1. Stater Feed Intake, ADG, and FE

The amount of starter feed intake, total DMI (milk DM + starter DM), and ADG were not influenced by supplemental FA source, starter CP content, or their interaction (Table 2). However, calves that received starter diets with LSO had a higher BW at weaning (*p* < 0.05) and end of study (*p* < 0.05) compared to those fed diets containing SBO. In addition, FE was improved in calves fed starters with LSO during the pre-weaning period (*p* < 0.05) and the entire period (*p* < 0.05) compared to those who received SBO. Experimental animals tended (*p* = 0.08) to have a heavier BW at weaning when fed starters with 22% CP compared to 18% CP. Moreover, FE was improved during pre-weaning (*p* < 0.05) and tended to be improved during the entire period of the experiment (*p* = 0.07) when calves were fed starters with 22% CP.

### 3.2. Skeletal Growth Items

The heart girth was higher (*p* < 0.05) at weaning and tended to be higher (*p* = 0.06) at the end of the experiment when calves received the n-3-22CP diet compared to other groups (Table 3). Hip height was the highest (*p* < 0.05) at the day of weaning and at the final measurement day of the experiment (*p* < 0.05) in calves fed the n-3-22CP diet compared to other experimental treatments. Wither height was higher in calves who received starters with LSO compared to calves who received SBO (*p* < 0.05).

### 3.3. Blood Chemistry, Insulin, and Inflammation Indicators

The concertation of blood glucose (*p* < 0.05) and insulin (*p* < 0.05) were highest in calves that received n-3-22CP diets among the experimental diets during the pre-weaning period (Table 4). However, compared to other diets, the highest BUN concentration was observed in calves fed the n-6-22CP diet during the pre-weaning (*p* < 0.05) and post-weaning (*p* < 0.05) periods. Calves fed LSO had a greater pre-weaning blood glucose concentration (*p* < 0.05) compared to those who received SBO. The lowest concentration of TNF-α (*p* < 0.05) in the blood during the pre-weaning period was observed in calves fed the n-3-22CP diet compared to other experimental groups. A lower concentration of SAA (*p* < 0.05) was observed in calves fed diets supplemented with LSO during the pre-weaning period compared to those who received SBO. The serum concentration of SAA tended to be lower (*p* = 0.07) when calves were fed starters with 22% CP content during the pre-weaning period. Starter FA source, CP content, and their interaction had no effect on blood HP concentration during the pre- and post-weaning periods (*p* > 0.05).

### 3.4. Urinary PD, MPS, and N Excretion

No interaction was found for supplemental FA and starter CP content with respect to urinary PD and MPS (*p* > 0.05; Table 5). However, the highest urinary N excretion was observed in calves fed n-6-22CP compared to other experimental diets (*p* < 0.05). Calves who received starter diets containing 22% CP had higher urinary allantoin (*p* < 0.05) and PD (*p* < 0.05) concentrations and MPS (*p* < 0.05) compared to those fed starter diets with 18% CP. Supplemental FA source had no effect on individual or total urinary PD excretions and MPS (*p* > 0.05). Calves who received diets containing LSO were shown to have less UN excretion (*p* < 0.05) than those fed diets with SBO.

## 4. Discussion

Neither supplemental FA source nor starter CP content influenced the starter intake, which could be largely because of relatively similar energy and CP level in starter diets. As indicated previously [25], diet energy and CP content can impact the appetite extent in farm animals. The amount of milk consumed was similar for all calves, meaning that it did not influence total DMI. It has been reported that some FA sources have a negative influence on starter intake in dairy calves during the early weeks of life [5,11]. This can be mainly due to interference with ruminal digestion of nutrients [26]. It seems that ruminal unprotected fat sources such as soybean oil have adverse effects on fermentation by microbes in calves [11]. In the current study, both SBO and LSO sources were ruminally bypassed through saponification of Ca-salt and thus negligible negative influence on microbial activity in the rumen and nutrient digestibility was expected to be observed by feeding the above supplemental FA [7,8].

The ADG was not changed among calves fed experimental diets, which is consistent with starter feed intake. However, weaning and final BWs were greater in calves fed LSO compared to SBO. In addition, feeding LSO improved FE, indicating that LSO supplementation was more efficient during the pre-weaning period. Improvement in N efficiency in calves fed LSO may contribute to improvement in FE in LSO-supplemented calves. In parallel with the improvement in FE and BW in calves fed LSO, wither height was also higher in these calves at weaning. The highest heart girth and hip height were observed in calves fed n-3-22CP, which might be attributed to more efficient N utilization and growth. A previous report indicated that certain individual FAs have the ability to increase N accretion in ruminants [12]. The current results are consistent with the latter study, which showed that LSO supplementation improved performance in young calves fed higher starter CP content, possibly through improving N utilization.

The highest concentrations of glucose and insulin in the blood were achieved in calves fed the n-3-22CP diet in the pre-weaning period. Higher concentrations of glucose in the blood can be an indicator of higher energy levels in dairy calves [27]. Greater insulin concentration, which is partly due to higher blood glucose levels during the pre-weaning period, can be a factor impacting the higher growth rate in young calves. As indicated in a previous study in young calves [28], a higher blood insulin concentration has the potential to increase amino acids (AA) absorption and hence improve growth rate. It is notable that this level of blood insulin is quite higher than the level causing the insulin-resistance condition that has been found in veal calves with higher milk feeding and lower solid feed intake [29]. Regardless of the starter protein content, LSO had a greater blood glucose concentration compared to SBO, probably due to higher feed efficiency achieved in calves who received the n-3 FA source. Considering the similar starter intake in calves fed SBO and LSO, greater blood glucose concentration in calves fed LSO may be related partly to improvement in immune function and more favorable conditions from the hormonal and metabolism perspectives.

The highest BUN concentration was detected in calves fed the n-6-22CP diet in the pre-weaning and post-weaning periods, indicating that in both periods, feeding high starter CP content along with SBO source cannot be as efficient as the n-3-22CP diet in young dairy calves. BUN has been considered an indicator of N efficiency in ruminants because it has a strong linear relationship with UN excretion [30]. Accordingly, the highest UN excretion was found in the n-6-22CP group. This indicates that in calves receiving 22% CP starter with SBO FA, extra N is excreted to the environment, but this was not the case in calves supplemented with LSO. Previous studies indicated that N retention was increased in dairy cows during early lactation [12], and in suckling lambs [14] when diets were supplemented with individual FAs. One study [12] reported that conjugated linoleic acid supplementation improved protein deposition in dairy cows during early lactation. The related mechanisms may be partly due to the pathways of gene expression [15] or anabolic hormones such as IGF-I [12], which could elevate N accretion. It appears that individual FAs have the potential for sparing energy in the way of N accretion rather than fat deposition [31,32]. In fact, it can be suggested that individual FAs may have different behavior with respect to N metabolism, and thus, various N efficiency and deposition through the body may be acquired when different FA sources are supplemented in diets. Our results showed that starter diets containing 22% CP that coincided with supplementation of LSO improved the efficiency of N utilization in young animals and this was not observed for SBO. Further research on organ and cell levels is required to explore the interaction of dietary CP content with FA on N utilization efficiency in growing animals.

The lowest and the highest concentration of TNF-α was observed in calves fed the n-3-22CP and n-6-18CP diets, respectively, indicating that both FA source and starter CP content can modulate the immune function of dairy calves. Moreover, lower SAA in calves fed diets containing LSO compared to those fed diets containing SBO suggested that supplementation of LSO improved the immune system in dairy calves. The anti-inflammatory properties of n-3 FA have been indicated in dairy calves [19,33]. For instance, linseed oil, which is rich in n-3 FA (α-linolenic acid), has been shown to have an anti-inflammatory impact on dairy calves [34,35]. Soybean oil, on the other side, which is rich in n-6 FA (linoleic acid), showed an inflammatory impact [11] and reduced the growth performance in suckling calves [4]. Some inflammatory properties of n-6 FA have been reported in young dairy calves [10]. Regardless of the FA effect on the immune function of dairy calves, higher starter CP content improved the immune function in calves, especially during the pre-weaning period. Others have shown that improved immune function in response to high dietary protein content is partly related to the role of some AAs, such as glutamine in the cytokines synthesis pathways (e.g., IL-1β and IL-6) in macrophages and monocytes, and phagocytosis and the production of reactive oxygen intermediates in rodent models and humans [36,37]. In addition, branched-chained AAs and methionine seem to play an important role in the immune function of dairy cows and calves [38]. Recently, some researchers evaluated the immune function of dairy calves fed starter diets containing different FAs [39,40]; however, more research is warranted to specifically evaluate the effect of starter CP level on immune function in suckling calves. Results of the current study suggest that LSO supplementation along with higher starter CP content in pre-weaned calves can probably be considered as a practical and on-farm method to reduce the inflammatory responses in pre-weaning calves, which are susceptible to infections in this critical period of time.

The results obtained for urinary PD and MPS indicate that the FA source did not have an impact on microbial activity, showing that saponification of both FA sources (linseed oil and soybean oil) as Ca-salt was efficient in preventing the negative effect of FA on ruminal fermentation. A recent study indicated that non-ruminal bypass of FA supplied through soybean oil negatively influenced microbial activity and dramatically reduced the MPS in dairy calves [4]. Because both FA sources bypassed the rumen, no difference was found in the urinary PD and MPS between FA sources in the current study. However, the highest UN excretion, which was found in the n-6-22CP group, was suggestive of the lowest N utilization efficiency in calves fed a high starter CP content supplemented with SBO. The higher starter CP content, however, increased the urinary PD excretion and hence improved the MPS, suggesting that greater N supplementation may stimulate the ruminal microbes to produce greater mass in young calves. This is in agreement with our previous report [20], where was shown that a higher starter CP content can provide more organic matter for ruminal microbes which consequently led to a higher MPS in young calves.

## 5. Conclusions

Compared to the Ca-salt of soybean oil (which contained a high level of n-6 FA), supplementation of Ca-salt of linseed oil (which contained a high level of n-3 FA) improved FE and some growth indices but reduced some inflammatory markers in suckling calves during the pre-weaning period. A marginal positive effect on growth items was found in calves fed starters with 22% CP compared to 18% CP. In conclusion, supplemental LSO in diets containing 22% CP increased the blood glucose and insulin concentrations and improved N efficiency and immune function of calves. The interaction of dietary individual FAs and CP content with respect to immune function and N efficiency needs to be further explored in future studies in young dairy calves.

## Figures and Tables

**Table 1 animals-13-00960-t001:** Experimental starter diets ingredients and chemical composition (g/kg of DM, unless otherwise stated).

	Experimental Diet ^1^			
	n-6 FA		n-3 FA	
Item	18CP	22CP	18CP	22CP
Ingredients, g/kg of DM				
Alfalfa hay, finely chopped	70	70	70	70
Barley grain, ground	110	110	110	110
Corn grain, coarsely ground	510	430	510	430
Soybean meal	230	310	230	310
Ca-salt of soybean oil (n-6 FA source)	30	30	0	0
Ca-salt of linseed oil (n-3 FA source)	0	0	30	30
Calcium carbonate	10	10	10	10
Di-calcium phosphate	5	5	5	5
Sodium bicarbonate	10	10	10	10
Salt	5	5	5	5
Vitamin and mineral mix ^2^	20	20	20	20
Chemical composition, (g/kg of DM, unless otherwise stated)				
Metabolizable energy ^3^, (Mcal/kg)	3.03	3.08	3.02	3.05
Crude protein	180	220	180	220
Neutral detergent fiber	187	182	187	182
Ether extract	49.7	49.5	49.7	49.5
Non-fiber carbohydrate ^4^	512	481	512	481
Calcium	90	90	90	90
Phosphorus	44	44	44	44

^1^ Experimental diets were; Starter supplemented with calcium salt of soybean oil (n-6 FA source) and containing 18% CP (n-6-18CP); Starter supplemented with calcium salt of soybean oil and (n-6 FA source) containing 22% CP (n-6-22CP); Starter supplemented with calcium salt of linseed oil (n-3 FA source) and containing 18% CP (n-3-18CP); Starter supplemented with calcium salt of linseed oil (n-3 FA source) and containing 22% CP (n-3-18CP). ^2^ Contained per kg of vitamin supplement: Vit A (IU) = 800,000, Vit D (IU) = 150,000, Vit E (IU) = 2000, Ca (g) = 110, P (g) = 40, Mg (g) = 50, Zn (mg) = 3000, Cu (mg) = 600, I (mg) = 150, Co (mg) = 130, Mn (mg) = 1700, Se (mg) = 100. ^3^ Calculated according to NRC (2001). ^4^ Non-fiber carbohydrate was calculated as DM − (NDF + CP + ether extract + ash) (NRC, 2001).

**Table 2 animals-13-00960-t002:** Least square means for starter intake, average daily gain, and feed efficiency in dairy calves fed different FA sources (Ca-salt of n-6 FA source vs. Ca-salt of n-3 FA source) and starter CP content (18 vs. 22%, DM basis) (*n* = 10 calves per treatment).

Item	Experimental Diet ^1^				SEM	*p-Value* ^2^		
	n-6 FA		n-3 FA			FA	CP	FA × CP
	18CP	22CP	18CP	22CP				
Starter feed intake, g/d								
Pre-weaning (d 3–63)	675	628	685	698	67.33	0.67	0.85	0.75
Post-weaning (d 64–83)	1937	2055	1965	1976	83.37	0.75	0.44	0.52
Entire period (d 3–83)	990	986	1005	1017	97.92	0.81	0.97	0.92
Milk intake, g/d	642	639	641	640	19.41	0.96	0.93	0.97
Total dry matter intake (milk + starter), g/d	1314	1266	1324	1335	95.82	0.67	0.85	0.74
Average daily gain, g/d								
Pre-weaning (d 3–63)	558	676	620	671	61.19	0.65	0.15	0.57
Post-weaning (d 64–83)	745	685	735	765	51.10	0.50	0.77	0.39
Entire period (d 3–83)	605	678	648	693	47.90	0.54	0.21	0.76
Body weight								
Initial (d 3)	40.0	40.8	40.5	39.8	1.22	0.80	0.96	0.46
Weaning (d 63)	73.5	75.9	77.7	80.1	1.84	0.02	0.08	0.94
Final (d 83)	88.4	89.7	92.4	95.3	1.75	0.05	0.15	0.54
Feed efficiency ^3^								
Pre-weaning (d 3–63)	0.42	0.53	0.51	0.59	0.04	0.04	0.05	0.71
Post-weaning (d 64–83)	0.39	0.34	0.38	0.41	0.03	0.45	0.57	0.21
Entire period (d 3–83)	0.40	0.48	0.48	0.54	0.03	0.05	0.07	0.95

^1^ Experimental diets were; Starter supplemented with calcium salt of soybean oil (n-6 FA source) and containing 18% CP (n-6-18CP); Starter supplemented with calcium salt of soybean oil and (n-6 FA source) containing 22% CP (n-6-22CP); Starter supplemented with calcium salt of linseed oil (n-3 FA source) and containing 18% CP (n-3-18CP); Starter supplemented with calcium salt of linseed oil (n-3 FA source) and containing 22% CP (n-3-18CP). ^2^ Statistical comparisons: FA = calcium salt of n-6 FA vs. calcium salt of n-3 FA; CP: 18% vs. 22% starter CP content (DM basis); FA × CP = interaction of supplemental FA sources and starter CP contents. ^3^ kg of body weight gain/kg of total dry matter intake.

**Table 3 animals-13-00960-t003:** Least square means for growth indices in dairy calves fed different FA sources (Ca-salt of n-6 FA source vs. Ca-salt of n-3 FA source) and starter CP content (18 vs. 22%, DM basis) (*n* = 10 calves per treatment).

Item	Experimental Diet ^1^				SEM	*p-Value* ^2^		
	n-6 FA		n-3 FA			FA	CP	FA × CP
	18CP	22CP	18CP	22CP				
Heart girth, cm								
d 63	97.1 ^b^	95.2 ^c^	94.4 ^c^	99.7 ^a^	0.98	0.59	0.31	0.03
d 83	100.4	101.0	99.3	105.7	1.04	0.46	0.8	0.06
Body length, cm								
d 63	49.3	49.1	49.5	50.7	0.68	0.42	0.65	0.53
d 83	53.9	52.0	52.9	54.8	0.63	0.41	0.94	0.10
Wither height, cm								
d 63	84.5	85.7	85.9	90.5	0.81	0.03	0.26	0.09
d 83	89.7	87.3	87.4	91.5	0.98	0.80	0.86	0.11
Hip height, cm								
d 63	89.6 ^b^	87.4 ^c^	87.6 ^c^	92.6 ^a^	0.83	0.25	0.32	0.02
d 83	89.8 ^c^	92.3 ^b^	90.0 ^c^	95.4 ^a^	1.09	0.23	0.29	0.01

^1^ Experimental diets were; Starter supplemented with calcium salt of soybean oil (n-6 FA source) and containing 18% CP (n-6-18CP); Starter supplemented with calcium salt of soybean oil and (n-6 FA source) containing 22% CP (n-6-22CP); Starter supplemented with calcium salt of linseed oil (n-3 FA source) and containing 18% CP (n-3-18CP); Starter supplemented with calcium salt of linseed oil (n-3 FA source) and containing 22% CP (n-3-18CP). ^2^ Statistical comparisons: FA = calcium salt of n-6 FA vs. calcium salt of n-3 FA; CP: 18% vs. 22% starter CP content (DM basis); FA × CP = interaction of supplemental FA sources and starter CP contents. ^a,b,c^ Values differ if they do not share a common letter (*p* < 0.05).

**Table 4 animals-13-00960-t004:** Least square means for blood metabolites in dairy calves fed different FA sources (Ca salt of n-6 FA source vs. Ca salt of n-3 FA source) and starter CP content (18 vs. 22%, DM basis) (*n* = 10 calves per treatment).

Item	Experimental Diet ^1^				SEM	*p-Value* ^2^		
	n-6 FA		n-3 FA			FA	CP	FA × CP
	18CP	22CP	18CP	22CP				
Glucose, mg/dL								
Pre-weaning	80.7 ^b^	77.2 ^c^	84.1 ^b^	98.7 ^a^	3.67	0.02	0.23	0.04
Post-weaning	58.1	63.5	72.4	67.2	4.19	0.06	0.97	0.27
BHB, mmol/L								
Pre-weaning	0.10	0.11	0.13	0.14	0.02	0.11	0.48	0.52
Post-weaning	0.19	0.21	0.29	0.22	0.04	0.09	0.39	0.13
Blood urea nitrogen, mg/dL								
Pre-weaning	21.5 ^b^	27.7 ^a^	19.8 ^c^	20.4 ^b^	1.04	0.01	0.02	0.03
Post-weaning	19.2 ^b^	25.3 ^a^	17.3 ^b^	16.5 ^c^	1.78	0.04	0.01	0.05
Insulin, IU/L								
Pre-weaning	5.87 ^b^	5.02 ^b^	4.84 ^c^	7.65 ^a^	0.34	0.05	0.03	0.01
Post-weaning	8.75	9.61	8.84	9.41	0.79	0.91	0.19	0.75
Inflammatory markers								
Tumor necrosis factor-α (pg/mL)								
Pre-weaning	488 ^a^	362 ^b^	377 ^b^	339 ^c^	14.09	0.02	0.01	0.04
Post-weaning	402	399	378	391	15.43	0.45	0.80	0.68
Serum amyloid A (mg/L)								
Pre-weaning	14.4	13.6	12.9	10.5	0.83	0.01	0.07	0.29
Post-weaning	13.5	11.7	11.9	13.0	0.79	0.89	0.16	0.78
Haptoglobin (μg/mL)								
Pre-weaning	12.0	12.2	12.5	11.3	0.60	0.81	0.52	0.37
Post-weaning	13.5	12.8	12.3	12.1	0.71	0.21	0.64	0.72

^1^ Experimental diets were; Starter supplemented with calcium salt of soybean oil (n-6 FA source) and containing 18% CP (n-6-18CP); Starter supplemented with calcium salt of soybean oil and (n-6 FA source) containing 22% CP (n-6-22CP); Starter supplemented with calcium salt of linseed oil (n-3 FA source) and containing 18% CP (n-3-18CP); Starter supplemented with calcium salt of linseed oil (n-3 FA source) and containing 22% CP (n-3-18CP). ^2^ Statistical comparisons: FA = calcium salt of n-6 FA vs. calcium salt of n-3 FA; CP: 18% vs. 22% starter CP content (DM basis); FA × CP = interaction of supplemental FA sources and starter CP contents. ^a,b,c^ Values differ if they do not share a common letter (*p* < 0.05).

**Table 5 animals-13-00960-t005:** Least square means for urinary purine derivatives and microbial protein synthesis in dairy calves fed different FA sources (Ca-salt of n-6 FA source vs. Ca-salt of n-3 FA source) and starter CP content (18 vs. 22%, DM basis) (*n* = 10 calves per treatment).

Item	Experimental Diet ^1^				SEM	*p-Value* ^2^		
	n-6 FA		n-3 FA			FA	CP	FA × CP
	18CP	22CP	18CP	22CP				
Allantoin, mmol/d	14.01	16.75	14.27	16.94	0.98	0.85	0.03	0.97
Uric acid, mmol/d Pre-weaning	1.39	1.39	1.32	1.26	0.05	0.45	0.82	0.85
Total PD, mmol/d	15.41	18.14	15.59	18.21	1.08	0.92	0.04	0.89
Microbial protein yield, g/d Pre-weaning	82.3	96.9	83.3	97.4	4.94	0.92	0.04	0.89
Urinary nitrogen, g/d	13.8 ^b^	16.9 ^a^	11.7 ^c^	12.3 ^b,c^	0.82	0.01	0.02	0.04

^1^ Experimental diets were; Starter supplemented with calcium salt of soybean oil (n-6 FA source) and containing 18% CP (n-6-18CP); Starter supplemented with calcium salt of soybean oil and (n-6 FA source) containing 22% CP (n-6-22CP); Starter supplemented with calcium salt of linseed oil (n-3 FA source) and containing 18% CP (n-3-18CP); Starter supplemented with calcium salt of linseed oil (n-3 FA source) and containing 22% CP (n-3-18CP). ^2^ Statistical comparisons: FA = calcium salt of n-6 FA vs. calcium salt of n-3 FA; CP: 18% vs. 22% starter CP content (DM basis); FA × CP = interaction of supplemental FA sources and starter CP contents. ^a,b,c^ Values differ if they do not share a common letter (*p* < 0.05).

## Data Availability

All data presented in the study were included in the manuscript.

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
