# Peer review of "Effects of Soybean and Linseed Oils Calcium Salts and Starter Protein Content on Growth Performance, Immune Response, and Nitrogen Utilization Efficiency in Holstein Dairy Calves"

_animals, 2023, doi:10.3390/ani13060960_

Round 1

Reviewer 1 Report

This study evaluated the effect of two different FAs and levels of CP content of starter on growth performance, immune function, and N efficiency in Holstein dairy calves. The study concept is clearly presented. The manuscript is well-written with nice explanations of the results. The authors provide a good introduction explaining the background and rationale for their study and their experimental objectives are clearly stated. I list below some comments for the authors’ consideration:

1) In the current study, authors used saponified soybean oil and linseed oil, which was rich in n-6 and n-3 FA, respectively, as FA sources treatment. If it is suitable to used n-6 and n-3 FA directly in the discussion part, because both oils have not only n-6 or n-3 FA but also other types.

2) What are the rumen by-pass rates of the two products?

3) As authors mentioned that n-3 FA could improve immune function of the dairy calves, if there was morbidity data to support this above hypothesis.

4) “less” was missing in Line 235.

5) There was no Table 4 in the manuscript.

Reviewer 2 Report

L207N ND 208: I was expecting a discussion about this (Calves 207 fed n-3 FA had greater pre-weaning blood glucose concentration (p < 0.05) compared to 208 those received n-6 FA) in the discussion section. 

Please explain in the discussion section your hypothesis about how n3 or n6 FA may have some effect on the metabolism of the level of CP used in the current experiment. 

Any discussion of how this FA early life supplementation may influence the response later in life of those animals? e.g. on rumen microbial community, performance, growth, milk yield, etc. 
